# Immune histories and natural infection protection during the omicron era
Hiam Chemaitelly [1,2] ✉, Houssein H. Ayoub [3], Niklas Bobrovitz [4,5], Peter Coyle [6,7,8], Patrick Tang[9], Mohammad R. Hasan[10], Hadi M. Yassine [7,11], Asmaa A. Al Thani[7,11], Zaina Al-Kanaani[6], Einas Al-Kuwari[6], Andrew Jeremijenko [6], Anvar Hassan Kaleeckal[6], Ali Nizar Latif[6], Riyazuddin Mohammad Shaik[6], Hanan F. Abdul-Rahim[12], Gheyath K. Nasrallah [7,11], Mohamed Ghaith Al-Kuwari[13,14], Hamad Eid Al-Romaihi[15], Mohamed H. Al-Thani[15], Abdullatif Al-Khal [6], Roberto Bertollini[15], Adeel A. Butt[2,6,16] & Laith J. Abu-Raddad [1,2,12,17] ✉

## Abstract

**Background** Past immunological events can either enhance or compromise an individual's future immune protection. This study investigated how different severe acute respiratory syndrome coronavirus 2 (SARS-CoV-2) natural infection histories before an omicron infection, with or without vaccination, influence protection against subsequent omicron reinfection. **Methods** Three national, matched, retrospective cohort studies were conducted in Qatar from February 28, 2020, to August 12, 2024 to compare incidence of omicron reinfection between individuals with two omicron infections (omicron double-infection cohort) and those with one (omicron single-infection cohort); the omicron double-infection cohort with individuals who had a pre-omicron infection followed by an omicron reinfection (pre-omicron-omicron double-infection cohort); and the pre-omicron-omicron double-infection cohort with the omicron single-infection cohort. **Results** Here we show that, in the first study, comparing the omicron double-infection cohort to the omicron single-infection cohort, the adjusted hazard ratio (aHR) is 1.27 (95% CI: 1.13–1.43); 0.93 (95% CI: 0.68–1.28) for the unvaccinated and 1.34 (95% CI: 1.18–1.52) for the vaccinated. In the second study, comparing the omicron double-infection cohort to the pre-omicron-omicron double-infection cohort, the aHR is 1.37 (95% CI: 1.13–1.65); 1.12 (95% CI: 0.63–1.97) for the unvaccinated and 1.42 (95% CI: 1.16–1.74) for the vaccinated. In the third study, comparing the pre-omicron-omicron double-infection cohort to the omicron single-infection cohort, the aHR is 0.97 (95% CI: 0.92–1.03); 0.75 (95% CI: 0.66–0.85) for the unvaccinated and 1.03 (95% CI: 0.97–1.09) for the vaccinated. **Conclusions** Immune history shapes protection against omicron reinfection, with pre-omicron-omicron immunity enhancing protection, while repeated similar exposures reduce protection against new variants.

## Plain language summary

This study explores how different histories of COVID-19 infection and vaccination affect protection against future infections. The investigators used data from Qatar's national health system, which includes information on COVID-19 tests, vaccinations, and hospitalisations. The study involved matching groups of people based on factors like age, sex, and vaccination status to compare their risk of reinfection. The study found that having two omicron infections provided less protection against future omicron reinfection than having just one omicron infection or a combination of pre-omicron and omicron infections. These results suggest that repeated infections with the same variant may not strengthen immunity as much as mixed exposures. The findings improve our understanding of how immune history shapes protection and may help guide vaccination and public health strategies.

Immune imprinting, broadly defined as the phenomenon in which a specific sequence of immunological events—arising from infection and/ or vaccination—can either enhance or compromise an individual's future immune protection against infection and severe disease, is an intriguing and debated concept, characterized by seemingly contradictory findings and inadequate exploration[1–5]. An indication of the limitations in understanding this phenomenon is the current inability to

confidently predict whether a particular immune history will provide superior or inferior protection against future infection. For instance, there has been intense discussion on how first-generation coronavirus disease 2019 (COVID-19) vaccines should be updated and whether a bivalent strategy (including both the ancestral and current viruses) or a monovalent strategy (including only current virus) should be adopted[6–10].

An understanding of how different immune histories affect protection against infection and severe disease can provide insights into the fundamental mechanisms of immunity, inform the development of effective vaccines, and optimize vaccine impact by tailoring strategies to specific populations[1–11]. This is particularly important in cases where an infection is already endemic, and the population has heterogeneous immune histories due to varying exposures to infection, variants, and vaccination[1]. Investigating how different immune histories affect protection can be approached at both molecular and population levels, as has been done in studies of severe acute respiratory syndrome coronavirus 2 (SARS-CoV-2) infection[1–5,12,13].

Building on this line of research, particularly in exploring immune history effects at the population level for hypothesis generation[4,5,12,13], we conducted three national, matched, retrospective cohort studies to examine how different natural infection histories, with or without vaccination, in addition to an omicron infection, may influence protection against subsequent omicron reinfection. The findings show that individuals with immunity from both pre-omicron and omicron infections, or from a single omicron infection, have greater protection against subsequent omicron reinfection than those with two omicron infections, suggesting that repeated exposure to the same immunological event may reduce protection against emerging variants.

## Methods

### Study population, data sources, and vaccination

This study comprised Qatar's resident population between February 28, 2020, the date of the first documented SARS-CoV-2-positive test, and August 12, 2024, the study's end date. Data on COVID-19 laboratory testing, vaccination, hospitalization, and death were sourced from Qatar's national integrated digital health information platform (Supplementary Methods). This platform captures all SARS-CoV-2-related data, including COVID-19 vaccinations, hospitalizations, and polymerase chain reaction (PCR) and medically supervised rapid-antigen tests, irrespective of location or facility (Supplementary Methods).

Qatar's testing program was extensive, with most infections identified through routine testing rather than symptomatic presentation (Supplementary Methods)[14,15]. COVID-19 vaccinations, comprising generally mRNA vaccines[16–18], were provided free of charge to all individuals, irrespective of citizenship status, exclusively through the public healthcare system[19]. The vaccination strategy prioritized frontline healthcare workers, individuals with severe or multiple chronic conditions, and individuals aged over 50 years[14]. Due to the very low coverage of omicron-based vaccines in Qatar[18], vaccination in this study predominantly refers to the original first-generation vaccines targeting the original virus.

Demographic data were obtained from the national health registry. Qatar's demographic structure is distinct. Only 9% of the population are over 50 years and 89% are resident expatriates from over 150 countries[20]. Further details on Qatar's population and COVID-19 databases have been previously published[12,14,15,20–23].

### Study design

Three national, matched, retrospective cohort studies were conducted, informed by our earlier cohort studies on Qatar's population[4,5,12,13,21,24–28], to investigate how different natural infection and vaccination histories before an omicron infection may affect protection against subsequent reinfection with omicron variants.

The first study compared the incidence of reinfection and of severe, critical, or fatal COVID-19 upon reinfection between the national cohort of individuals with a primary omicron infection followed by an omicron reinfection (omicron double-infection cohort) and the national cohort of individuals with only a primary omicron infection (omicron single-infection cohort) (Fig. 1a).

The second study compared these outcomes between the omicron double-infection cohort and the national cohort of individuals with a primary pre-omicron infection followed by an omicron reinfection (pre-omicron-omicron double-infection cohort) (Fig. 1b). The third study compared these outcomes between the pre-omicron-omicron double-infection cohort and the omicron single-infection cohort (Fig. 1c).

Incidence of reinfection was defined as a SARS-CoV-2-positive test after the start of follow-up. Severity of reinfections was determined by trained medical personnel independent of study investigators, based on individual chart reviews. These assessments followed the World Health Organization's (WHO) guidelines for defining COVID-19 severity[29], criticality[29], and fatality[30] (Supplementary Methods)[31].

### Cohorts' matching

Cohorts were matched exactly one-to-one by sex, 10-year age group, nationality, number of coexisting conditions (Supplementary Methods), number of vaccine doses, and vaccine type, as ascertained at the start of follow-up. Matching was also performed based on the testing method (PCR versus rapid-antigen testing), reason for testing, and calendar week of the SARS-CoV-2 test that defined reinfection for the omicron double-infection cohort, reinfection for the pre-omicron-omicron double-infection cohort, and primary infection for the omicron single-infection cohort. Matching by calendar time ensured that matched pairs were concurrently present in Qatar. This comprehensive matching strategy, informed by prior studies on Qatar's population[4,5,12,13,21,24–28], ensured balance of confounders across study cohorts.

Iterative matching was implemented to ensure that, at the start of follow-up, individuals were alive and had the same vaccine type and number of doses as their match. Further details can be found in Supplementary Methods.

### Cohorts' follow-up

Follow-up started from 90 days after the reinfection for individuals in the omicron double-infection cohort in the studies comparing this cohort to either the omicron single-infection cohort or the pre-omicron-omicron double-infection cohort. Similarly, follow-up was from 90 days after the reinfection for individuals in the pre-omicron-omicron double-infection cohort in the study comparing this cohort to the omicron single-infection cohort.

The 90-day cutoff, consistent with the conventional definition of SARS-CoV-2 reinfection as a documented infection occurring ≥90 days after a previous infection, avoids misclassifying prolonged test positivity as reinfection[32–35].

For exchangeability[21,36], both members of each matched pair were censored at the earliest occurrence of receiving an additional vaccine dose. Accordingly, individuals were followed until the first of any of the following events: a documented SARS-CoV-2 reinfection (irrespective of symptoms), a new vaccine dose (with matched-pair censoring), death, or the administrative end of follow-up at the end of the study.

### Variant ascertainment

The duration of dominance of each SARS-CoV-2 variant (Supplementary Fig. 1) was determined based on Qatar's variant genomic surveillance[37–39]. This surveillance program comprised viral genome sequencing[37] and multiplex real-time reverse-transcription PCR (RT-qPCR) variant screening[38] performed on weekly collected random positive clinical samples (Supplementary Methods).

### Ethics

The institutional review boards at Hamad Medical Corporation (reference number: MRC–01–20–1078) and Weill Cornell Medicine–Qatar (reference number: 20–00017) approved this retrospective study with a waiver of informed consent. The study was reported according to the Strengthening the Reporting of Observational Studies in Epidemiology (STROBE; Supplementary Table 1).

### Statistical analysis

Matched cohorts were characterized through descriptive statistics and compared using standardized mean differences (SMDs). An SMD of ≤0.1

indicated adequate matching[40]. Cumulative incidence of reinfection, defined as proportion of individuals at risk with reinfection as a primary endpoint during follow-up, was estimated using the Kaplan–Meier estimator method.

Cumulative incidence was reported unadjusted to provide a direct and intuitive representation of the observed data, and adjusted estimates were also provided in a sensitivity analysis. Schoenfeld residuals and log-log plots

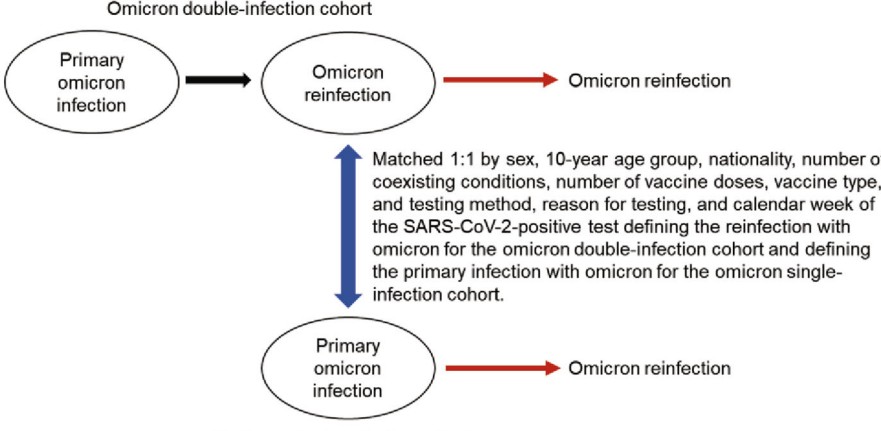

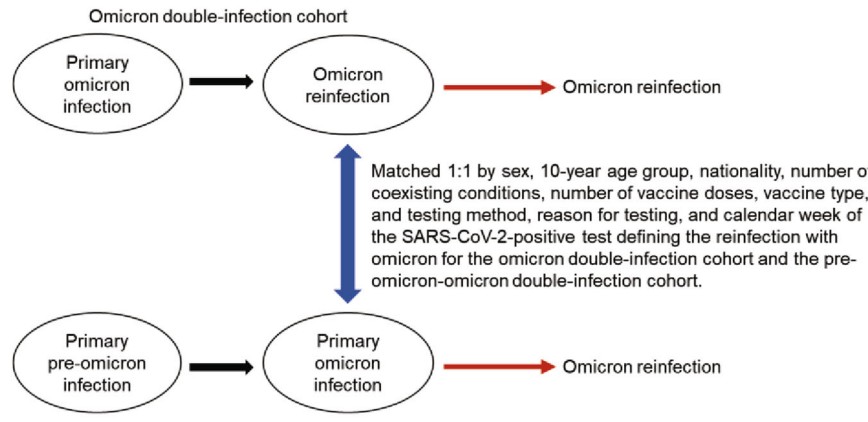

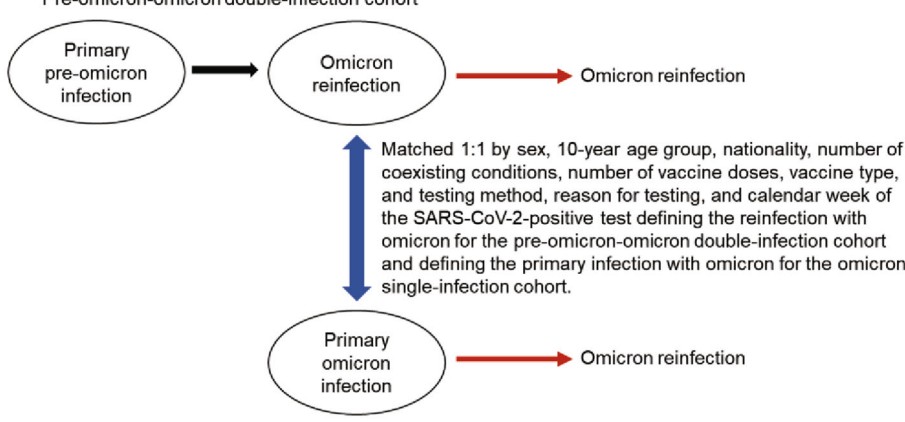

**Fig. 1 | Study design. Schematic diagram illustrating the study design for comparing the incidence of SARS-CoV-2 reinfection across study cohorts.** Study design for the analysis comparing incidence of reinfection between **a** the omicron double-infection cohort and the omicron single-infection cohort, **b** the omicron double-infection cohort and the pre-omicron-omicron double-infection cohort, and **c** the pre-omicron-omicron double-infection cohort and the omicron single-infection cohort.

for survival curves were used to examine the proportional-hazards assumption.

Incidence rate of reinfection, defined as number of individuals with reinfection in a cohort divided by number of person-weeks contributed by all individuals in the cohort, was estimated, with the associated 95% confidence interval (CI), using a Poisson log-likelihood regression model.

Adjusted hazard ratios (aHRs) comparing the incidence of reinfection between study cohorts, along with the corresponding 95% CIs, were calculated using Cox regression models. These models were adjusted for matching factors to ensure unbiased estimation of the standard variance[41], as well as for SARS-CoV-2 testing rates. To explore differences in the risk of reinfection over time, aHRs were also estimated by 6-month intervals from the start of follow-up using separate Cox regressions, with failure (reinfection) restricted to specific time intervals.

A subgroup analysis was performed by estimating the aHRs stratified by vaccination status. CIs were not adjusted for multiplicity, and interactions were not investigated, except in sensitivity analysis.

The conceptual approach of this study involved constructing matched cohorts from distinct full-cohort samples in a way that allowed their disaggregation into separate substudies for unvaccinated and vaccinated individuals, enabling the assessment of study outcomes within these subgroups. To confirm these results, a sensitivity analysis was conducted, where the same study outcomes for these subgroups were calculated using interaction terms between study cohorts and vaccination status. Cox interaction models were applied to the full cohorts to evaluate these interactions.

Statistical analyses were performed using Stata/SE version 18.0 (Stata Corporation, College Station, TX, USA).

### Reporting summary

Further information on research design is available in the Nature Portfolio Reporting Summary linked to this article.

## Results

### Omicron double-infection cohort and omicron single-infection cohort

Figure 1a schematizes the study design. Supplementary Fig. 2 details the cohorts' selection process. Supplementary Data 1 describes the matched cohorts' baseline characteristics, each comprising 11,185 individuals. Supplementary Fig. 3A depicts the median dates of immunological events occurring before or after the start of follow-up among individuals who experienced these events within these analyzed cohorts.

The median duration between the two infections that define the omicron double-infection cohort was 309 days (interquartile range [IQR]: 239–474 days). Among vaccinated individuals, the median date of the first vaccine dose was March 22, 2021 (IQR: February 23, 2021–May 20, 2021) in the omicron double-infection cohort and April 7, 2021 (IQR: February 28, 2021–May 30, 2021) in the omicron single-infection cohort.

During follow-up, 675 reinfections were documented in the omicron double-infection cohort and 498 in the omicron single-infection cohort (Supplementary Fig. 2). Supplementary Table 2 presents the distribution of these reinfections by omicron subvariant. None of these reinfections progressed to severe, critical, or fatal COVID-19. Cumulative incidence of reinfection was 8.9% (95% CI: 8.2–9.6%) for the omicron double-infection cohort and 6.5% (95% CI: 6.0–7.2%) for the omicron single-infection cohort, after 660 days of follow-up (Fig. 2a and Supplementary Data 2). Supplementary Fig. 4A provides the cumulative incidence of reinfection stratified by vaccination status.

The aHR comparing the incidence of reinfection in the omicron double-infection cohort to that in the omicron single-infection cohort was 1.27 (95% CI: 1.13–1.43; Table 1a) and appeared stable over the follow-up period (Supplementary Fig. 5A). In the subgroup analysis by vaccination status, the aHR was 0.93 (95% CI: 0.68–1.28) for the unvaccinated group and 1.34 (95% CI: 1.18–1.52) for the vaccinated group (Fig. 3a, Supplementary Table 3A, and Supplementary Data 3). The aHR was similar for those who

received only primary-series vaccination and those who received three or more doses (Supplementary Table 3A).

### Omicron double-infection cohort and pre-omicron-omicron double-infection cohort

Figure 1b schematizes the study design. Supplementary Fig. 6 details the cohorts' selection process. Supplementary Data 1 describes the matched cohorts' baseline characteristics, each comprising 3573 individuals. Supplementary Fig. 3B depicts the median dates of immunological events occurring before or after the start of follow-up among individuals who experienced these events within these analyzed cohorts.

The median duration between the two infections that define the omicron double-infection cohort was 282 days (IQR: 222–455 days). The median duration between the two infections that define the pre-omicron-omicron double-infection cohort was 698 days (IQR: 521–876 days). Among vaccinated individuals, the median date of the first vaccine dose was March 23, 2021 (IQR: February 24, 2021–May 19, 2021) in the omicron double-infection cohort and April 20, 2021 (IQR: March 4, 2021–June 3, 2021) in the pre-omicron-omicron double-infection cohort.

During follow-up, 268 reinfections were documented in the omicron double-infection cohort and 185 in the pre-omicron-omicron double-infection cohort (Supplementary Fig. 6). Supplementary Table 2 presents the distribution of these reinfections by omicron subvariant. None of these reinfections progressed to severe, critical, or fatal COVID-19. Cumulative incidence of reinfection was 10.1% (95% CI: 9.0–11.3%) for the omicron double-infection cohort and 7.0% (95% CI: 6.1–8.1%) for the pre-omicron-omicron double-infection cohort, after 660 days of follow-up (Fig. 2b and Supplementary Data 2). Supplementary Fig. 4B provides the cumulative incidence of reinfection stratified by vaccination status.

The aHR comparing incidence of reinfection in the omicron double-infection cohort to that in the pre-omicron-omicron double-infection cohort was 1.37 (95% CI: 1.13–1.65; Table 1b) and appeared stable over the follow-up period (Supplementary Fig. 5B). In the subgroup analysis by vaccination status, the aHR was 1.12 (95% CI: 0.63–1.97) for the unvaccinated group and 1.42 (95% CI: 1.16–1.74) for the vaccinated group (Fig. 3b, Supplementary Table 3B, and Supplementary Data 3). The aHR was similar for those who received only primary-series vaccination and those who received three or more doses (Supplementary Table 3B).

### Pre-omicron-omicron double-infection cohort and omicron single-infection cohort

Figure 1c schematizes the study design. Supplementary Fig. 7 details the cohorts' selection process. Supplementary Data 1 describes the matched cohorts' baseline characteristics, each comprising 37,771 individuals. Supplementary Fig. 3C depicts the median dates of immunological events occurring before or after the start of follow-up among individuals who experienced these events within these analyzed cohorts.

The median duration between the two infections that define the pre-omicron-omicron double-infection cohort was 435 days (IQR: 292–567 days). Among vaccinated individuals, the median date of the first vaccine dose was April 27, 2021 (IQR: March 14, 2021–May 31, 2021) in the pre-omicron-omicron double-infection cohort and April 13, 2021 (IQR: March 8, 2021–May 27, 2021) in the omicron single-infection cohort.

During follow-up, 2784 reinfections were documented in the pre-omicron-omicron double-infection cohort and 2726 in the omicron single-infection cohort (Supplementary Fig. 7). Supplementary Table 2 presents the distribution of these reinfections by omicron subvariant. None of these reinfections progressed to severe, critical, or fatal COVID-19. Cumulative incidence of reinfection was 8.9% (95% CI: 8.5–9.2%) for the pre-omicron-omicron double-infection cohort and 8.7% (95% CI: 8.4–9.0%) for the omicron single-infection cohort, after 840 days of follow-up (Fig. 2c and Supplementary Data 2). Supplementary Fig. 4C provides the cumulative incidence of reinfection stratified by vaccination status.

The aHR comparing incidence of reinfection in the pre-omicron-omicron double-infection cohort to that in the omicron single-infection

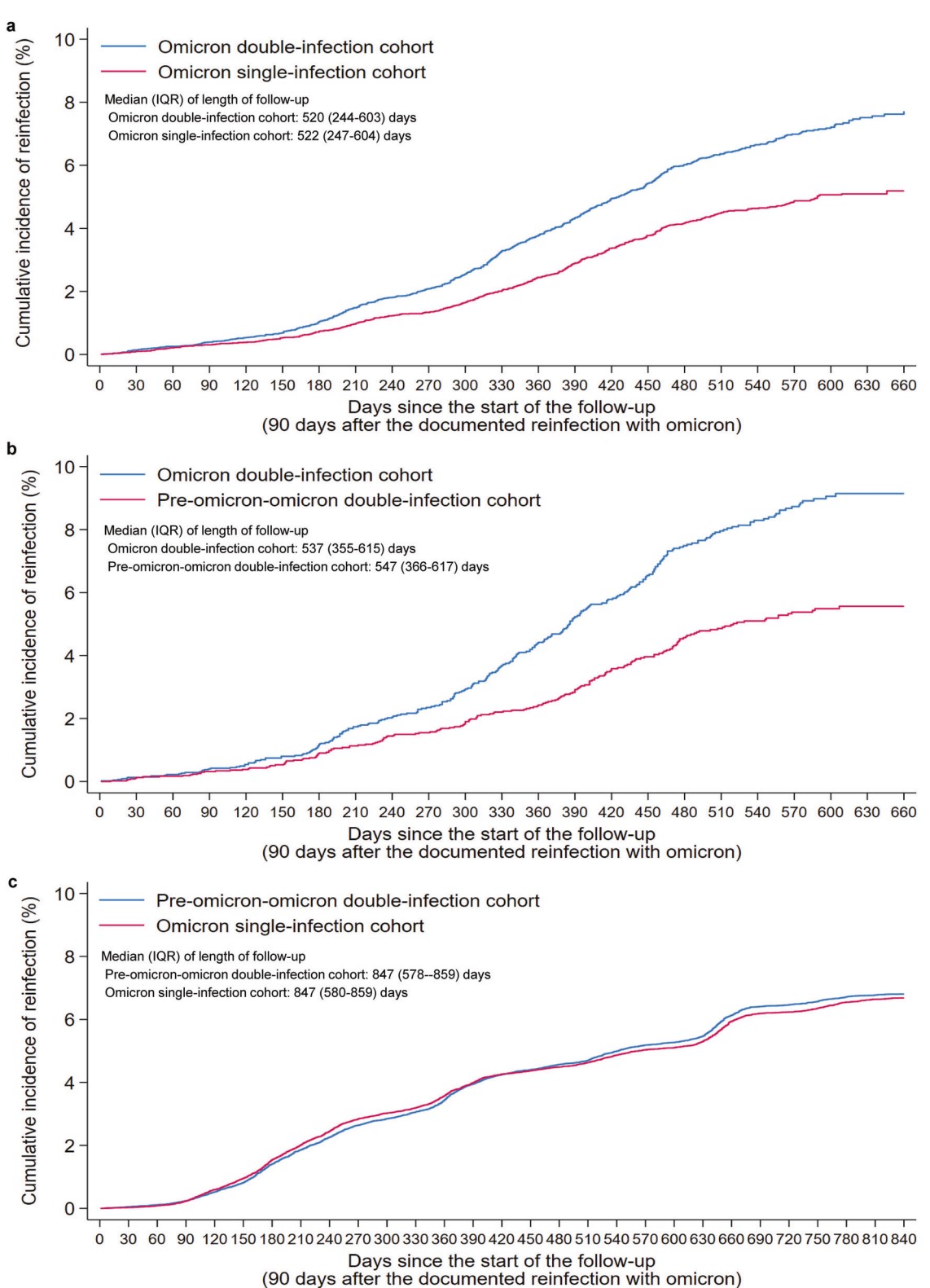

**Fig. 2 | Incidence of SARS-CoV-2 reinfection.** Cumulative incidence in the studies comparing the incidence of SARS-CoV-2 reinfection between **a** the omicron double-infection cohort and the omicron single-infection cohort, **b** the omicron double-infection cohort and the pre-omicron-omicron double-infection cohort, and **c** the pre-omicron-omicron double-infection cohort and the omicron single-infection cohort.

cohort was 0.97 (95% CI: 0.92–1.03; Table 1c) and appeared stable over the follow-up period (Supplementary Fig. 5C). In the subgroup analysis by vaccination status, the aHR was 0.75 (95% CI: 0.66–0.85) for the unvaccinated group and 1.03 (95% CI: 0.97–1.09) for the vaccinated group (Fig. 3c, Supplementary Table 3C, and Supplementary Data 3). The aHR was similar for those who received only primary-series vaccination and those who received three or more doses (Supplementary Table 3C).

**Table 1 | Hazard ratios for the incidence of SARS-CoV-2 reinfection**

| Epidemiological measure | Exposure cohort | Control cohort |
|---|---|---|
| a. Omicron double-infection cohort (Exposure cohort) versus omicron single-infection cohort (Control cohort)[a] | | |
| Sample size | 11,185 | 11,185 |
| Total follow-up time (person-weeks) | 693,544 | 698,929 |
| Incident infections | 675 | 498 |
| Incidence rate of reinfection (per 10,000 person-weeks; 95% CI) | 9.7 (9.0–10.5) | 7.1 (6.5–7.8) |
| Unadjusted hazard ratio for SARS-CoV-2 reinfection (95% CI) | 1.37 (1.22–1.53) | |
| Adjusted hazard ratio for SARS-CoV-2 reinfection (95% CI)[b] | 1.27 (1.13–1.43) | |
| b. Omicron double-infection cohort (Exposure cohort) versus pre-omicron-omicron double-infection cohort (Control cohort)[c] | | |
| Sample size | 3357 | 3357 |
| Total follow-up time (person-weeks) | 235,240 | 237,570 |
| Incident infections | 268 | 185 |
| Incidence rate of reinfection (per 10,000 person-weeks) | 11.4 (10.1–12.8) | 7.8 (6.7–9.0) |
| Unadjusted hazard ratio for SARS-CoV-2 reinfection (95% CI) | 1.47 (1.22–1.78) | |
| Adjusted hazard ratio for SARS-CoV-2 reinfection (95% CI)[d] | 1.37 (1.13–1.65) | |
| c. Pre-omicron-omicron double-infection cohort (Exposure cohort) versus omicron single-infection cohort (Control cohort)[e] | | |
| Sample size | 37,771 | 37,771 |
| Total follow-up time (person-weeks) | 3,573,409 | 3,572,390 |
| Incident infections | 2784 | 2726 |
| Incidence rate of reinfection (per 10,000 person-weeks) | 7.8 (7.5–8.1) | 7.6 (7.4–7.9) |
| Unadjusted hazard ratio for SARS-CoV-2 reinfection (95% CI) | 1.02 (0.97–1.08) | |
| Adjusted hazard ratio for SARS-CoV-2 reinfection (95% CI)[f] | 0.97 (0.92–1.03) | |

Hazard ratios comparing the incidence of SARS-CoV-2 reinfection between a the omicron double-infection cohort and the omicron single-infection cohort, b the omicron double-infection cohort and the pre-omicron-omicron double-infection cohort, and c the pre-omicron-omicron double-infection cohort and the omicron single-infection cohort.

CI confidence interval, PCR polymerase chain reaction, SARS-CoV-2 severe acute respiratory syndrome coronavirus 2.

[a]Persons in the omicron double-infection cohort were matched exactly one-to-one to persons in the omicron single-infection cohort by sex, 10-year age group, nationality, number of coexisting conditions, number of vaccine doses, vaccine type, in addition to testing method (PCR versus rapid-antigen testing), reason for testing, and calendar week of the SARS-CoV-2-positive test defining the reinfection with omicron for the omicron double-infection cohort and defining the primary infection with omicron for the omicron single-infection cohort. Each matched pair was followed from 90 days after the date of the reinfection for the individual in the omicron double-infection cohort.

[b]Adjusted for sex, 10-year age group, nationality, number of coexisting conditions, number of vaccine doses, vaccine type, testing method, reason for testing, and calendar week of the SARS-CoV-2-positive test defining the reinfection with omicron for the omicron double-infection cohort and defining the primary infection with omicron for the omicron single-infection cohort, and testing rate.

[c]Persons in the omicron double-infection cohort were matched exactly one-to-one to persons in the pre-omicron-omicron double-infection cohort by sex, 10-year age group, nationality, number of coexisting conditions, number of vaccine doses, vaccine type, in addition to testing method (PCR versus rapid-antigen testing), reason for testing, and calendar week of the SARS-CoV-2-positive test defining the reinfection with omicron for the omicron double-infection cohort and the pre-omicron-omicron double-infection cohort. Each matched pair was followed from 90 days after the date of the reinfection for the individual in the omicron double-infection cohort.

[d]Adjusted for sex, 10-year age group, nationality, number of coexisting conditions, number of vaccine doses, vaccine type, testing method, reason for testing, and calendar week of the SARS-CoV-2-positive test defining the reinfection with omicron for the omicron double-infection cohort and for the pre-omicron-omicron double-infection cohort, and testing rate.

[e]Persons in the pre-omicron-omicron double-infection cohort were matched exactly one-to-one to persons in the omicron single-infection cohort by sex, 10-year age group, nationality, number of coexisting conditions, number of vaccine doses, vaccine type, in addition to testing method (PCR versus rapid-antigen testing), reason for testing, and calendar week of the SARS-CoV-2-positive test defining the reinfection with omicron for the pre-omicron-omicron double-infection cohort and defining the primary infection with omicron for the omicron single-infection cohort. Each matched pair was followed from 90 days after the date of the reinfection for the individual in the pre-omicron-omicron double-infection cohort.

[f]Adjusted for sex, 10-year age group, nationality, number of coexisting conditions, number of vaccine doses, vaccine type, testing method, reason for testing, and calendar week of the SARS-CoV-2-positive test defining the reinfection with omicron for the pre-omicron-omicron double-infection cohort and defining the primary infection with omicron for the omicron single-infection cohort, and testing rate.

### Additional analyses

The sensitivity analysis generating the *adjusted* cumulative incidence (Supplementary Fig. 8) produced incidence curves that closely resembled those observed in the main unadjusted analysis.

The sensitivity analysis, which estimated study outcomes for unvaccinated and vaccinated subgroups using interaction terms, confirmed similar results (Supplementary Fig. 9) and demonstrated evidence of interaction, with effects being modified by vaccination status ($p$-value < 0.001 for all three cohort analyses).

### Discussion

Although the most recent immunological event in the investigated cohorts was an omicron infection, the incidence of subsequent infections varied based on prior infection histories (Fig. 2) and differed between vaccinated and unvaccinated subgroups (Fig. 3). The varying levels of protection associated with different immunological histories may be linked to the memory component of the immune response[4,10,12], considering that antibody-mediated protection from vaccines and natural infections tends to wane rapidly[11,14,27,42–47]. These findings underscore the need for further investigation into these phenomena, including controlled experimental studies both in vitro and in vivo, to elucidate the biological mechanisms underlying the observed population-level effects.

The identified immune history effects at the population level complement those observed earlier[4,5,12]. In previous research, we found that a hybrid of pre-omicron and omicron immunity, whether from infection or vaccination, was associated with enhanced protection against future omicron reinfection, possibly due to the broadening of the immune response, exposure to more immunological events, and higher antibody titers[4,5]. We also found that repeated sequential immunological events of the same kind were associated with reduced protection against a new type of immunological event[4,12].

The study findings overall align with these general patterns. A history of two sequential omicron infections after vaccination was associated with reduced protection compared to having only one omicron infection following vaccination (Fig. 3a). Additionally, two sequential omicron infections after vaccination conferred less protection than one omicron infection following an earlier pre-omicron infection and vaccination (Fig. 3b). A single omicron infection following vaccination appeared to provide comparable protection to that of one omicron infection following an earlier pre-omicron infection and vaccination (Fig. 3c). Finally, consistent with our previous study[5], an omicron infection after a pre-omicron infection, in the absence of vaccination, offered greater protection than a history of only one omicron infection (Fig. 3c).

This study focused on the impact of immune history on protection against infection, but the nature of the variant challenge can also influence these effects. For example, our earlier studies suggested that specific immune history effects were more pronounced when the infection challenge involved the BA.5 variant[4,5,12]. Differences in the protection conferred by different immune histories often appear to result from a mismatch between specific immune memory and a substantially different immune challenge[4,10,12]. However, due to the size of the study cohorts, it was not possible to stratify the analysis by the dominant variant at each time point. As a result, the specific role of the variant challenge in shaping protection against infection remains an area for further investigation. Additionally, the cohort size restricted our ability to examine how the timing or the order of a sequence of immunological events impacts protection against infection and contributes to the observed effects.

Although we intended to investigate the impact of different immune histories on protection against severe COVID-19 outcomes, the lack of severe cases among the study cohorts rendered this analysis infeasible. Although some COVID-19 patients were hospitalized, none met the WHO criteria for severe or critical COVID-19, and there were no COVID-19-related deaths after the longitudinal review of their individual charts. This outcome aligns with the generally milder nature of omicron variants[31,48,49], the more robust and slowly waning protection against severe disease conferred by natural infection and vaccination compared to protection against infection[11,12,14,15,27,42–44,47,50,51], and the

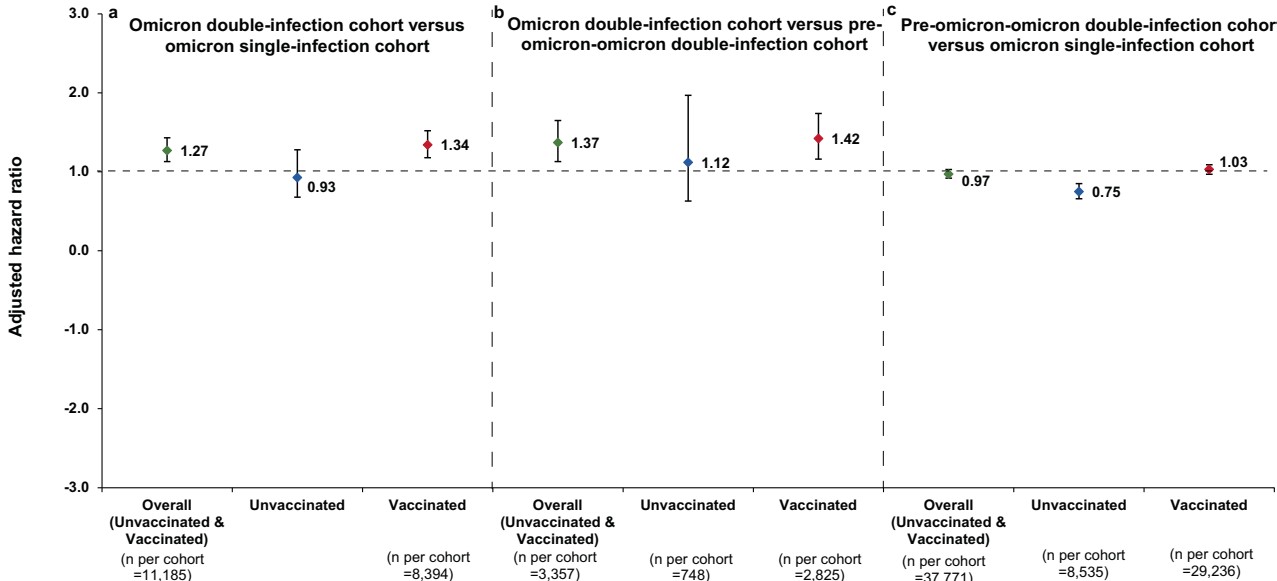

**Fig. 3 | Subgroup analysis. SARS-CoV-2 reinfection risk, stratified by vaccination status.** Hazard ratios for the incidence of SARS-CoV-2 reinfection in the studies comparing the incidence of SARS-CoV-2 reinfection between **a** omicron double-infection cohort and the omicron single-infection cohort, **b** the omicron double-infection cohort and the pre-omicron-omicron double-infection cohort, and **c** the pre-omicron-omicron double-infection cohort and the omicron single-infection cohort. Error bars indicate the corresponding 95% confidence intervals.

possibility that viral mutations optimizing immune escape may be associated with reduced virulence[52,53].

This study has limitations. As an observational study, the investigated cohorts were neither blinded nor randomized, making it not possible to rule out unmeasured or uncontrolled confounding. Since the compared cohorts had different immune histories—some more protective than others—conditioning on these histories may introduce a collider bias[54,55]. Furthermore, the timing between prior exposures varied across the studied cohorts in the three analyses, potentially serving as an additional marker of exposure not fully accounted for by the covariates matched and adjusted for in this study.

A thorough examination of potential bias in this type of study design for assessing immune history effects, particularly the possibility of collider bias, suggests that such bias is not likely to have an appreciable impact or explain the observed results[56]. This conclusion is supported by the use of rigorous exact matching to balance infection risk across cohorts, aimed at equalizing the propensity for infection and informed by knowledge of SARS-CoV-2 epidemiology within this specific population[56]. Moreover, study estimates were adjusted for testing frequency, a key factor influencing variations in the propensity for documented infection[56]. The presented analyses are also part of a broader series investigating immune history effects[4,5,12,13], and the overall findings from this series are inconsistent with the presence of an appreciable collider bias[56]. Instead, the results support the conclusion that, if such bias exists, its impact is marginal and mitigated by the study design.

The rigorous matching algorithm employed in this study ensures comparability between the study cohorts by accounting for observable factors that could influence infection risk[20,57-60]. While this approach enhances internal validity, it may limit generalizability due to the reduced cohort size. The smaller cohort size also restricts the ability to conduct additional subgroup analyses or precisely estimate effects, such as those based on the number of vaccine doses or vaccine types.

Each cohort was compared to the other two in separate analyses, leading to slight variations in cohort composition due to the matching process. These differences affected cohort demographics, follow-up times, and cumulative incidence rates across comparisons. For instance, the cumulative incidence in the omicron double-infection cohort differed between the two comparisons (8.9% versus 10.1%) due to variations in cohort composition and follow-up times.

Although the matching process accounted for key infection risk factors[20,57-60], data limitations prevented matching on geography or occupation. Certain populations may have specific occupational risks that were not adjusted for, such as healthcare workers, who were also prioritized for vaccination over other population groups. However, healthcare workers constitute a very small proportion of Qatar's population. Moreover, Qatar is essentially a city-state where nationality, age, and sex serve as effective proxies for socio-economic status and occupation[20,57-60].

It is not likely that health conditions, weaker immune systems, or genetic factors could account for the results, as the matching process included multiple coexisting conditions and nationality, the latter possibly serving as a partial proxy for genetic factors. Furthermore, Qatar's population is predominantly young and largely consists of healthy, working-age migrants, with only a very small proportion of elderly individuals who are more likely to have weaker immune systems[14,20,23]. The observed effects were sizable in magnitude in comparison, making it unlikely that these factors could explain the findings.

Given Qatar's relatively young and healthy population[20,23], the study findings may not be generalizable to countries with a larger elderly or comorbid population. Documented SARS-CoV-2 infections do not capture all infections that have occurred within the population. However, since under-ascertainment likely affected the study cohorts similarly, the impact on the study estimates—based on relative comparisons between the cohorts—may be mitigated.

The study has strengths. It was conducted on a national scale, encompassing a diverse population with a range of national backgrounds, and leveraged extensive, validated databases established through numerous SARS-CoV-2 infection studies. Exact matching was employed to ensure rigorous pairing of individuals across the cohorts. The matching methodology had been validated in previous studies with various epidemiological designs and tested with control groups to confirm null effects[14,17,19,42,61]. These studies demonstrated that this approach effectively controls for differences in infection exposure[14,17,19,42,61].

In conclusion, protection against subsequent omicron reinfection appears to vary based on prior infection and vaccination histories. The findings support the concept that immunity involving both pre-omicron and omicron exposures provides enhanced protection, whereas repeated sequential exposures to the same immunological event reduce protection

against new variants. These findings highlight the need for further investigation into the biological mechanisms underlying the observed differences in protection against infection. Controlled experimental studies are essential to better understand how different immunological histories shape immune responses to future variants.

## Data availability

The dataset of this study is a property of the Qatar Ministry of Public Health that was provided to the researchers through a restricted-access agreement that prevents sharing the dataset with a third party or publicly. The data are available under restricted access for preservation of confidentiality of patient data. Access can be obtained through a direct application for data access to His Excellency the Minister of Public Health (https:// https://emsfsa.moph. gov.qa/en/Pages/eservices.aspx). The raw data are protected and are not available due to data privacy laws. Requests for access are assessed by the Ministry of Public Health in Qatar, and approval is granted at its discretion. In compliance with data privacy laws and the data-sharing agreement with the Ministry of Public Health in Qatar, no datasets, whether raw or de-identified, can be publicly released by the researchers. Aggregate data are available within the paper and its supplementary information. Source data for figures and tables can be found in Supplementary Data 1-3.

## Code availability

Standard epidemiological analyses were conducted using standard commands in STATA/SE 18.0. These commands have been published at: https://github.com/IDEGWCMQ/Cohort/blob/main/Cohort-Code.do and https://doi.org/10.5281/zenodo.15615967[62].

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

## Acknowledgements

We acknowledge the many dedicated individuals at Hamad Medical Corporation, the Ministry of Public Health, the Primary Health Care Corporation, Qatar Biobank, Sidra Medicine, and Weill Cornell Medicine-Qatar for their diligent efforts and contributions to make this study possible. The authors are grateful for institutional salary support from the Biomedical Research Program and the Biostatistics, Epidemiology, and Biomathematics Research Core, both at Weill Cornell Medicine-Qatar, as well as for institutional salary support provided by the Ministry of Public Health, Hamad Medical Corporation, and Sidra Medicine. The authors are also grateful for the Qatar Genome Programme and Qatar University Biomedical Research Center for institutional support for the reagents needed for the viral genome sequencing. HC gratefully acknowledges salary support from the Junior Faculty Transition to Independence Program at Weill Cornell Medicine–Qatar and L'Oréal-UNESCO For Women In Science Middle East Regional Young Talents Program. Statements made herein are solely the responsibility of the authors. ChatGPT was exclusively utilized to verify grammar and refine the English phrasing in our text. No other functionalities or applications of ChatGPT were employed beyond this specific scope. Following the use of this tool, the authors thoroughly reviewed and edited the content as necessary and take full responsibility for the accuracy and quality of the publication.

## Author contributions

H.C. co-designed the study, performed the statistical analyses, and co-wrote the first draft of the article. L.J.A. conceived and co-designed the study, led the statistical analyses, and co-wrote the first draft of the article. H.C. and L.J.A. accessed and verified all the data. P.C. designed mass PCR testing to allow routine capture of variants and conducted viral genome sequencing. P.T. and M.R.H. designed and conducted multiplex, RT-qPCR variant screening and viral genome sequencing. H.M.Y. and A.A.A.T. conducted viral genome sequencing. All authors (H.C., H.H.A., N.B., P.C., P.T., M.R.H., H.M.Y., A.A.A.T., Z.A.-K., E.A.-K., A.J., A.H.K., A.N.L., R.M.S., H.F.A.-R., G.K.N., M.H.A.-K., H.E.A.-R., M.H.A.-T., A.A.-K., R.B., A.A.B. and L.J.A.) contributed to data collection and acquisition,

database development, discussion and interpretation of the results, and to the writing of the article. All authors have read and approved the final manuscript.

## Competing interests

The authors declare the following competing interests: A.A.B. has received institutional grant funding from Gilead Sciences unrelated to the work presented in this paper. Otherwise, the authors declare no competing interests.

## Additional information

[1]Infectious Disease Epidemiology Group, Weill Cornell Medicine-Qatar, Cornell University, Doha, Qatar. [2]Department of Population Health Sciences, Weill Cornell Medicine, Cornell University, New York, NY, USA. [3]Mathematics Program, Department of Mathematics and Statistics, College of Arts and Sciences, Qatar University, Doha, Qatar. [4]Department of Emergency Medicine, Cumming School of Medicine, University of Calgary, Calgary, AB, Canada. [5]Centre for Health Informatics, Cumming School of Medicine, University of Calgary, Calgary, AB, Canada. [6]Hamad Medical Corporation, Doha, Qatar. [7]Department of Biomedical Science, College of Health Sciences, QU Health, Qatar University, Doha, Qatar. [8]Wellcome-Wolfson Institute for Experimental Medicine, Queens University, Belfast, UK. [9]Department of Pathology and Laboratory Medicine, University of British Columbia, Vancouver, BC, Canada. [10]Department of Pathology and Molecular Medicine, McMaster University, Hamilton, ON, Canada. [11]Biomedical Research Center, QU Health, Qatar University, Doha, Qatar. [12]Department of Public Health, College of Health Sciences, QU Health, Qatar University, Doha, Qatar. [13]Primary Health Care Corporation, Doha, Qatar. [14]College of Medicine, Qatar University, Doha, Qatar. [15]Ministry of Public Health, Doha, Qatar. [16]Department of Medicine, Weill Cornell Medicine, Cornell University, New York, NY, USA. [17]College of Health and Life Sciences, Hamad bin Khalifa University, Doha, Qatar. ✉e-mail: hsc2001@qatar-med.cornell.edu; lja2002@qatar-med.cornell.edu

