## [Transparent Peer Review file · Communications Medicine]

Immune Histories and Natural Infection Protection during the Omicron Era

Corresponding Author: Professor Laith Abu-Raddad

Version 0:

Reviewer comments:

Reviewer #1

(Remarks to the Author)

1. This study has major strengths including a large sample size with a high-level of complete data collection on vaccinations and infections and potential confounding factors, and the ability to apply rigorous matching on several important variables that makes it credible that the estimated associations represent causal effects of the exposure-contrasts being studied. A particular strength is the matching tightly controls the timing of the most recent Omicron infection, and the clear use of 'index time' that renders the associations being assessed clearly interpretable in terms of temporality.
2. The authors note a limitation was that the Omicron sub-variant of the exposing virus may impact results, yet the analyses did not/could not account for this factor. I wonder if it would be possible to provide some information on the Omicron sub-variants causing Omicron re-infections, for example showing data on estimated sub-variant frequencies over time based on a sequence data base such as GISAID, in the context of the follow-up of the different groups being studied.
3. Was there evidence that any of the three associations/effects being studied changed over time?
4. The statistical analysis plan did not include interaction tests for whether the studied exposure-associations/effects were modified by vaccinated vs. unvaccinated status. Yet in the discussion, it is indicated that this work addresses the objective about how immune history that depends on sequence of infections and vaccination impact risk of Omicron re-infection. Given vaccination is an immunity-conferring event, that this study has high-quality data on vaccination status, and the sample size is large, I do not understand the rationale for excluding formal hypothesis testing for interaction/effect modification. Figure 3 suggests the results depend on vaccination status. Could a rationale for not including interaction tests be supplied?
5. While the study has rigorous strengths as noted above, the nature of the first and third exposure effects being studied (A and C) have a particular susceptibility to bias in that 2 prior infections vs. 1 prior infection over a similar time frame could itself be a marker of exposure that is not necessarily controlled for by the set of covariates matched for and adjusted for. This issue is also relevant for exposure effect B given that the 2 Omicron prior infections must tend to happen over a shorter duration of time than the 1 pre-Omicron followed by an Omicron infection. The manuscript appropriately acknowledges the possibility of bias, yet it does not consider this specific source of bias; please consider whether discussing this specific issue explicitly would add value. It may also be relevant to consider a sensitivity analysis to quantify how results depend on potential biases caused by this issue. In the Discussion the authors cite several papers supporting rigor of the matching/analysis approach; perhaps it would be possible to apply some of these arguments to the present potential problem, e.g., with elaboration in supplemental material.
6. For groups with two prior infections, it would add value to communicate the distribution of the intervals of time between these two infections, as it is a component of immune history.
7. As noted above a strength is solid confounding control through matching and covariate-adjustment. One potentially confounding covariate that seemed missing was timing of vaccination, as temporality of immunity-conferring events impacts protection. Number of vaccine doses and vaccine type were controlled for, but not timing of vaccination, if I understood the statistical methods correctly.

8. The Cox modeling analyses adjust for covariates, but the cumulative incidence analyses (via Kaplan-Meier curves) do not. What is the rationale for not also adjusting the cumulative incidence curves for the same covariates adjusted for with the Cox modeling? One common way to do this is to estimate $E_X[P(T \leq t|A=a, X)]$ where X are covariates and A is the binary exposure variable (a is the particular group) being considered. The authors point out that it was necessary to use a covariate-adjusted version of the Cox model to 'ensure unbiased estimation of the standard variance, as well as for SARS-CoV-2 testing rates.' Are there similar issues for cumulative incidence analysis? Regardless, obtaining as complete as possible confounding control for the cumulative incidence analyses is clearly of value.

9. Showing Fig. 2 results stratified by vaccinated vs. unvaccinated would be useful, given the significance of the vaccinated vs. unvaccinated covariate.

Reviewer #2

(Remarks to the Author)

Dr. Chemaitelly and colleagues calculated the adjusted hazard ratios for reinfection with a SARS-CoV-2 omicron variant based on different immune backgrounds with the aim to find evidence of immune imprinting effects at the population level. Relevant data from digital health records from Qatar's national and universal public healthcare system were made available for this study. Their findings on increased infection rates due to repeated homologous challenge support immune imprinting. The manuscript is well written. The study design and methodology are clear and transparent. The results are of interest to guide fundamental immune studies, vaccine development and vaccine policies.

General comments

1. The use of the term 'hybrid immunity' to describe immunity against pre-omicron and omicron infections can be confusing for some readers. Hybrid immunity is often used to describe immunity induced by a combination of vaccination and infection. Consider rephrasing throughout the manuscript.
2. The authors use the term 'priming' in relation to the first documented SARS-CoV-2 infection. Although this term correctly applies to immunity towards SARS-CoV-2 antigens other than Spike, many (probably even most) have been primed (and boosted) with ancestral spike-based vaccines prior to SARS-CoV-2 infection. As priming of Spike-specific immunity (hence the term original antigenic sin) and the antigenic evolution of Spike (driven by pre-existing population immunity) are critical components of immune imprinting the use of this term needs careful consideration. Please clarify this in the text and consider using different terminology for the first encountered infection throughout the manuscript.
3. Related to my previous comment and potential collider bias based on the conditioning of histories; it would be of added value to perform subgroup analysis (or adjust groups in the analysis performed in Figure 3 and Supplementary Table 1) on vaccine-primed (ancestral strain) versus infection-primed individuals in these cohorts.

Specific comments

1. Ease of interpretation of Figure 3 would benefit from i) making it more narrow so the data is closer to each other ii) depicting the aHR and CI for the entire group for comparison, iii) including a horizontal (dotted) line shown at aHR=1.0 to visualize significance of the observed differences and iv) noting the sample sizes for each group.
2. It is mentioned that risk populations were prioritized for vaccination in Qatar, which includes co-existing conditions (which is adjusted for), but also specific occupational risks such as healthcare workers (not adjusted for). Could these combined factors influence findings in Figure 3 and Supplementary table 1?
3. Could the authors comment on the difference in incidence of omicron infection and aHR in the double omicron primed cohorts A vs B (8.9% vs 10.1%)?
4. P16, line 338: In the author contributions section, does PVC refer to Dr. Peter Coyle? If so, maybe the author's name misses an initial.
5. The supplementary text on the matching procedure (page 11, line 206-209) is not clear to me. Is the data from specific individuals from one study arm (single omicron infection) recycled in the second arm (dual omicron infection) for the calculation of the aHR of omicron infection?

Reviewer #3

(Remarks to the Author)

In the present study Abu-Raddad and colleagues conducted an epidemiological study demonstrating the incidence of SARS-CoV-2 omicron (O) infections among individuals with different histories of SARS-CoV-2 immune challenges. In particular, they conducted pairwise comparisons of three groups of individuals: individuals previously infected by O, individuals with two previous O infections, and individuals who got infected by pre-omicron (preO) variant followed by O infection. They also did a subgroup analysis where they further separated and compared individuals that did or did not receive wild-type-(WT)-based vaccine.

The manuscript includes important epidemiological data with a robust grouping approach where individuals are matched 1:1 by various possibly confounding factors. While the observed differences between the groups seem reliable, the interpretation in the context of immunological imprinting is not appropriate for several reasons:

- 1) Immune imprinting shapes the immune response to infection when the infection is already established and affects the severity of the outcome but is unlikely to impact protection from infection. All these individuals had a previous O infection and have successfully recovered from it. Also, no differences in the severity of previous O infections were reported for different groups. This means that they generated an immune response that is able to neutralize O (whether this response was imprinted by previous exposures or not) and have a certain degree of protection from reinfection with O (which varies with

different immune histories regardless of imprinting).

2) Even if imprinting would impact protection from infection its effects should be pronounced in the case of multiple previous exposures to WT and pre-omicron antigens that are structurally more distinct from O antigens than antigens from other O subvariants. Moreover, multiple previous exposures to O antigens should offer better protection against severe O infection, first because the immune response recognizing O is repeatedly boosted and second because there is more chance for overriding the imprinting by WT vaccine and preO variants. The manuscript claims the opposite namely that individuals with WT(vaccine)-O(infection) and those with WT(vaccine)-preO(infection)-O(infection) exposure history are better protected from O infection than individuals with WT(vaccine)-O(infection)- O(infection) exposure history. In the case of the last comparison where unvaccinated individuals with preO(infection)-O(infection) exposure history were found to be better protected than those with only O(infection) this again disproves the role of imprinting since previous exposure to antigenically distinct preO variants should imprint the response to O infection, while O infection cannot imprint response to O infection (if we disregard the genetic differences of omicron subvariants that were also not accounted for in this study). This difference is more likely due to additional immune events and higher antibody titers as many studies demonstrated low adaptive immunity following SARS-CoV-2 infection without previous immune challenges.

I have a feeling that the authors do not completely understand the mechanism of immunological imprinting and the difference between immunological imprinting and original antigenic sin. Immunological imprinting occurs when the immune response against a certain pathogen, usually a virus, cannot adapt to the mutations found in a new viral variant but rather remains locked in an initial clonal repertoire imprinted by the previous exposure to a similar yet structurally distinct viral variant. The boosting of adaptive immunity recognizing both variants, however, still occurs and usually offers sufficient protection from new variants. In rare cases, an imprinted immune response can lead to a failure of control over viral replication if a virus mutates to the point where it is still recognized but no longer efficiently neutralized (due to weaker binding of antibodies and TCRs to mutated regions) by the imprinted adaptive immune response. Only then we can talk about the original antigenic sin, and such cases were previously described for influenza and dengue virus infection.

Overall, I find the results of this study important but I recommend the authors to interpret their findings in a different way since the proposed immunological imprinting model does not fit.

Major comments:

- 1) How were the SARS-CoV-2 variants determined? By sequencing?
- 2) Wow was the timing between immune challenges for different groups. Particularly important is to see the average time that passed between the last exposure and the period of observation.
- 3) The differences between groups are observed only in the case of vaccinated individuals for the first two comparisons and only in the case of unvaccinated individuals for the third comparison. How do authors interpret this? Please add to the discussion.

Minor comments:

- 4) Line 53,75: Antigenic sin is a negative term and cannot enhance immune protection. Immune imprinting and antigenic sin are also not synonyms, please look at the definition above.
- 5) Line 55: Imprinting effects of what on what?
- 6) Line 67: Hybrid immunity in the context of SARS-CoV-2 infection is usually used to describe immunity by vaccination and infection; this study was not designed to compare that.
- 7) The results show that individuals who have fewer past omicron infections are less likely to become infected by omicron again. Might there be a genetic or some other kind of predisposition such as exposure, or a bad immune system, making some individuals more susceptible to omicron infection?
- 8) Lines 144-146: This study was designed to only compare differences depending on prior histories of infection and not vaccination (you cannot compare unvaccinated with vaccinated because there is no matching anymore)
- 9) Lines 146-147: Different immune histories impact the protection from O reinfection also without the imprinting.
- 10) Line 146: For clarity, please add that the only omicron-infected group was also unvaccinated.
- 11) Were the subvariants or at least the timing of infections comparable between the groups? It would be helpful to add a graph depicting the distribution of infection time points for each group.
- 12) Line 179: Is precluded the right word here?
- 13) Why the aHR for unratified data is not presented in Figure 3 and why stratified data by vaccination is not presented in Figure 2?
- 14) Why are results from Figure 2 not commented on in the discussion?

Version 1:

Reviewer comments:

Reviewer #1

(Remarks to the Author)

A thorough and effective revision in response to all the reviewer comments.

Reviewer #2

(Remarks to the Author)

I thank the authors for thoroughly addressing my questions, comments and suggestions where possible. Regarding the suggested subgroup analyses on vaccine-primed vs infection-primed individuals, I understand that despite the substantial study cohort sizes, the possibility to perform these analyses was limited and I acknowledge the effort. Therefore, I do not have any further questions or comments regarding this manuscript.

Reviewer #3

(Remarks to the Author)

The authors have successfully addressed most of my points and the manuscript has improved considerably. However, a few points still need to be addressed:

1) The comment regarding the timing of immune challenges has not been adequately addressed. Please provide the median dates of all infections and vaccinations (or the median duration between immune challenges, but please be consistent). Particularly important is the median time between the last immune challenge and the detected omicron reinfection during the observation period. This may help in the interpretation of your results. It would be helpful if you could graph the timing of the immune challenges (median dates) as a timeline.

2) Line 222-224: Protection against SARS-CoV-2 infection is neither strong nor long-lasting, there is little protection as individuals are re-infected repeatedly, even within a few weeks after vaccination when protection should be at its peak. Even if you wanted to talk about protection against severe infection, I would be cautious about saying that it is strong or durable (if you have found studies that show this, please cite them). Less severe infections are also due to the virus losing its fitness while adopting mutations to escape immune control.

Version 2:

Reviewer comments:

Reviewer #2

(Remarks to the Author)

The authors have addressed all reviewer comments effectively and thoroughly.

Reviewer #3

(Remarks to the Author)

Authors have addressed all my concerns.

Immune Histories and Natural Infection Protection during the Omicron Era: A Population-Based Cohort Analysis

REPLY TO REVIEWERS' COMMENTS

We are grateful to the editor, editorial board, and reviewers for assessing our work and for their insightful and useful feedback and suggestions. Please find below a point-by-point reply addressing each of the comments. We have also incorporated these suggestions in the revised manuscript, as noted below. We would be pleased to address any additional matters, should that be necessary.

Note: All references to the revised manuscript pertain to the marked copies of the manuscript files including changes implemented through "track changes".

Referees' comments:

Referee #1 (Remarks to the Author):

- 1. This study has major strengths including a large sample size with a high-level of complete data collection on vaccinations and infections and potential confounding factors, and the ability to apply rigorous matching on several important variables that makes it credible that the estimated associations represent causal effects of the exposure-contrasts being studied. A particular strength is the matching tightly controls the timing of the most recent Omicron infection, and the clear use of 'index time' that renders the associations being assessed clearly interpretable in terms of temporality.*

Comment: We thank the reviewer for the time and effort put into this review, the assessment of our work, and the constructive feedback on our manuscript that enriched it and improved its readability. Please find below a point-by-point reply addressing each of the reviewer's comments.

- 2. The authors note a limitation was that the Omicron sub-variant of the exposing virus may impact results, yet the analyses did not/could not account for this factor. I wonder if it would be possible to provide some information on the Omicron sub-variants causing Omicron re-infections, for example showing data on estimated sub-variant frequencies over time based on a sequence data base such as GISAID, in the context of the follow-up of the different groups being studied.*

Answer: Thank you for the excellent suggestion. To address the reviewer's comment, we have added a table showing the distribution of omicron subvariants among the reinfections in each cohort study (Supplementary Table 1) and cited this table in the main text (Results, Page 5, Paragraph 4; Page 6, Paragraph 5; and Page 7, Paragraph 5).

Additionally, we have included a figure illustrating SARS-CoV-2 infection incidence and variants in Qatar from February 5, 2020, the onset of the pandemic, to August 12, 2024, the end of the study (Supplementary Fig. 8). Furthermore, we have provided further details on Qatar's variant genomic surveillance program, which includes viral genome sequencing and multiplex real-time reverse-transcription PCR (RT-qPCR) variant screening performed on weekly collected random positive clinical samples^{1,2} (Methods, Page 16, Paragraph 5 and Page 17, Paragraph 1).

3. Was there evidence that any of the three associations/effects being studied changed over time?

Answer: Although the cohorts are large enough to assess the overall associations, they are not large enough to examine associations at different times of follow-up, as the 95% confidence intervals overlap. To address the reviewer's comment, we have now added the analysis by time of follow-up to demonstrate this point (Supplementary Fig. 3).

This additional analysis is now discussed and reported in the Methods (Methods, Page 18, Paragraph 1), Results (Results, Page 6, Paragraph 2; Page 7, Paragraph 2; and Page 8, Paragraph 2), and Supplementary Appendix (Supplementary Fig. 3).

4. The statistical analysis plan did not include interaction tests for whether the studied exposure-associations/effects were modified by vaccinated vs. unvaccinated status. Yet in the discussion, it is indicated that this work addresses the objective about how immune history that depends on sequence of infections and vaccination impact risk of Omicron re-infection. Given vaccination is an immunity-conferring event, that this study has high-quality data on vaccination status, and the sample size is large, I do not understand the rationale for excluding formal hypothesis testing for interaction/effect modification. Figure 3 suggests the results depend on vaccination status. Could a rationale for not including interaction tests be supplied?

Answer: Excellent point, thank you. The conceptual approach of this study involved constructing matched cohorts in a way that allowed their disaggregation into separate sub-studies for unvaccinated and vaccinated individuals, enabling the assessment of study outcomes within these subgroups. An alternative analytical approach, as suggested by the reviewer, is to derive these effects using interaction terms. Both approaches should yield similar results.

To address the reviewer's comment, a sensitivity analysis has now been included, in which the same study outcomes for these subgroups were calculated using interaction terms between study cohorts and vaccination status. Cox interaction models were applied to the full cohorts to evaluate these interactions. The analysis confirmed similar results (Supplementary Fig. 7) and demonstrated evidence of interaction, with effects being modified by vaccination status (p-value < 0.001 for all three cohort analyses).

This sensitivity analysis is now discussed and reported in the Methods (Methods, Page 18, Paragraph 3), Results (Results, Page 8, Paragraph 4), and Supplementary Appendix (Supplementary Fig. 7).

5. While the study has rigorous strengths as noted above, the nature of the first and third exposure effects being studied (A and C) have a particular susceptibility to bias in that 2 prior infections vs. 1 prior infection over a similar time frame could itself be a marker of exposure that is not necessarily controlled for by the set of covariates matched for and adjusted for. This issue is also relevant for exposure effect B given that the 2 Omicron prior infections must tend to happen over a shorter duration of time than the 1 pre-Omicron followed by an Omicron infection. The manuscript appropriately acknowledges the possibility of bias, yet it does not consider this specific source of bias; please consider whether discussing this specific issue explicitly would add value. It may also be relevant to consider a sensitivity analysis to quantify how results depend on potential biases caused by this issue. In the Discussion the authors cite several papers supporting rigor of the matching/analysis approach; perhaps it would be possible to apply some of these arguments to the present potential problem, e.g., with elaboration in supplemental material.

Answer: We agree with the reviewer regarding this potential source of bias. This issue was addressed in our discussion under the umbrella term 'collider bias,' which has been thoroughly investigated in a previous publication³. For this reason, we mentioned it briefly and cited the relevant literature.

In response to the reviewer's comment, we have now expanded our discussion of this issue, linking it to our earlier related analyses and results, which support the conclusion that, if such bias exists, its impact is likely minimal and mitigated by the study design (Discussion, Page 11, Paragraphs 1 and 2).

6. For groups with two prior infections, it would add value to communicate the distribution of the intervals of time between these two infections, as it is a component of immune history.

Answer: Thank you for the useful suggestion. This addition has now been included in the Results section (Results, Page 5, Paragraph 3; Page 6, Paragraph 4; and Page 7, Paragraph 4).

7. As noted above a strength is solid confounding control through matching and covariate-adjustment. One potentially confounding covariate that seemed missing was timing of vaccination, as temporality of immunity-conferring events impacts protection. Number of vaccine doses and vaccine type were controlled for, but not timing of vaccination, if I understood the statistical methods correctly.

Answer: Yes, indeed, this is correct, and we agree with the reviewer's point. The reason this criterion was not included in the matching process is that it would have substantially reduced the size of the cohorts due to the matching attrition. However, its exclusion did not introduce appreciable differences between the cohorts, as matching by the number of vaccine doses and vaccine type (indirectly) ensured comparability. The dates of vaccine doses were largely similar between the two cohorts in each analysis, making this additional matching criterion unnecessary.

In response to the reviewer's comment, we have now explicitly reported the median date and interquartile range (IQR) of the first vaccine dose in each cohort, demonstrating the similarity in dates even though this criterion was not part of the matching process. This addition has now been included in the Results section (Results, Page 5, Paragraph 3; Page 6, Paragraph 4; and Page 7, Paragraph 4).

8. The Cox modeling analyses adjust for covariates, but the cumulative incidence analyses (via Kaplan-Meier curves) do not. What is the rationale for not also adjusting the cumulative incidence curves for the same covariates adjusted for with the Cox modeling? One common way to do this is to estimate $E_X[P(T \leq t|A=a, X)]$ where X are covariates and A is the binary exposure variable (a is the particular group) being considered. The authors point out that it was necessary to use a covariate-adjusted version of the Cox model to 'ensure unbiased estimation of the standard variance, as well as for SARS-CoV-2 testing rates.' Are there similar issues for cumulative incidence analysis? Regardless, obtaining as complete as possible confounding control for the cumulative incidence analyses is clearly of value.

Answer: The reason we did not adjust for covariates in the cumulative incidence analyses is that, by study design, the cohorts were matched on these covariates. Essentially, the matching process inherently "adjusted" the cumulative incidence curves. Furthermore, this approach provided a direct and intuitive representation of the actual observed data in these cohorts. However, in the Cox regressions, the additional adjustment was necessary to address a specific technical requirement for ensuring unbiased estimation of the standard variance.

In response to the reviewer's comment, we have now included the adjusted cumulative incidence analyses as an additional sensitivity analysis, which yielded results consistent with the unadjusted analyses (Supplementary Fig. 6). This sensitivity analysis is also now discussed and reported in the Methods (Methods, Page 17, Paragraph 3), Results (Results, Page 8, Paragraph 3), and Supplementary Appendix (Supplementary Fig. 6).

9. Showing Fig. 2 results stratified by vaccinated vs. unvaccinated would be useful, given the significance of the vaccinated vs. unvaccinated covariate.

Answer: Thank you for the useful suggestion. This figure has now been added as suggested in the Supplementary Appendix (Supplementary Fig. 2) and cited in the Results section (Results, Page 6, Paragraph 1; Page 7, Paragraph 1; and Page 8, Paragraph 1).

Referee #2 (Remarks to the Author):

*Dr. Chemaitelly and colleagues calculated the adjusted hazard ratios for reinfection with a SARS-CoV-2 omicron variant based on different immune backgrounds with the aim to find evidence of immune imprinting effects at the population level. Relevant data from digital health records from Qatar's national and universal public healthcare system were made available for this study. Their findings on increased infection rates due to repeated homologous challenge support immune imprinting.
The manuscript is well written. The study design and methodology are clear and transparent.*

The results are of interest to guide fundamental immune studies, vaccine development and vaccine policies.

Comment: We thank the reviewer for the time and effort put into this review, the assessment of our work, and the constructive feedback on our manuscript that enriched it and improved its readability. Please find below a point-by-point reply addressing each of the reviewer's comments.

General comments

1. The use of the term 'hybrid immunity' to describe immunity against pre-omicron and omicron infections can be confusing for some readers. Hybrid immunity is often used to describe immunity induced by a combination of vaccination and infection. Consider rephrasing throughout the manuscript.

Answer: Thank you for the useful point. We have revised the term 'hybrid immunity' to 'pre-omicron and omicron immunity' throughout the manuscript and Supplementary Appendix (Multiple instances throughout the manuscript and Supplementary Appendix).

2. The authors use the term 'priming' in relation to the first documented SARS-CoV-2 infection. Although this term correctly applies to immunity towards SARS-CoV-2 antigens other than Spike, many (probably even most) have been primed (and boosted) with ancestral spike-based vaccines prior to SARS-CoV-2 infection. As priming of Spike-specific immunity (hence the term original antigenic sin) and the antigenic evolution of Spike (driven by pre-existing population immunity) are critical components of immune imprinting the use of this term needs careful consideration. Please clarify this in the text and consider using different terminology for the first encountered infection throughout the manuscript.

Answer: Excellent point, thank you. We have revised the names of these cohorts by replacing the term 'primed' with 'infection' throughout the manuscript and Supplementary Appendix to avoid confusion regarding the intended meaning of the term 'primed' (Multiple instances throughout the manuscript and Supplementary Appendix).

3. Related to my previous comment and potential collider bias based on the conditioning of histories; it would be of added value to perform subgroup analysis (or adjust groups in the analysis performed in Figure 3 and Supplementary Table 1) on vaccine-primed (ancestral strain) versus infection-primed individuals in these cohorts.

Answer: Thank you for the insightful suggestion. We agree with the reviewer that such an analysis is of great interest. We attempted to conduct this subgroup analysis, but it was only feasible for the C study comparison, comparing the pre-omicron-omicron double-infection cohort versus the omicron single-infection cohort. The results suggested differences between being vaccine-primed versus infection-primed (aHR of 1.28 (95% CI: 1.12–1.46) for vaccine-primed and 0.98 (95% CI: 0.92–1.05) for infection-primed).

The analysis was not feasible in the other two comparisons, as vaccination nearly always occurred before the first omicron infection in the omicron double-infection cohort. In summary,

the current study, given its specific design, is not suited to investigate the effect of vaccine versus infection priming due to statistical underpowering, as such analysis is a subgroup analysis of a subgroup analysis. Addressing the question of differences between vaccine priming and infection priming—specifically, the effect of the order of a sequence of immunological events—would be better suited to a study specifically designed to examine this effect and address the caveats of such comparison.

To address this comment, we have now clarified that this study is underpowered to assess the extent to which the specific order of immunological events influences protection against infection (Discussion, Page 10, Paragraph 2).

Specific comments

1. Ease of interpretation of Figure 3 would benefit from i) making it more narrow so the data is closer to each other ii) depicting the aHR and CI for the entire group for comparison, iii) including a horizontal (dotted) line shown at aHR=1.0 to visualize significance of the observed differences and iv) noting the sample sizes for each group.

Answer: Excellent and useful suggestions, thank you. These have been implemented accordingly (Fig. 3).

2. It is mentioned that risk populations were prioritized for vaccination in Qatar, which includes co-existing conditions (which is adjusted for), but also specific occupational risks such as healthcare workers (not adjusted for). Could these combined factors influence findings in Figure 3 and Supplementary table 1?

Answer: This is not likely. Healthcare workers represent a very small proportion of Qatar's population and, consequently, only a minimal number of healthcare workers should have contributed to these cohorts. Moreover, nationality serves as an effective proxy for socio-economic status and occupation in Qatar⁴⁻⁸, and matching by nationality should have partially adjusted for any occupational effects. This point has now been clarified in the revised manuscript (Discussion, Page 12, Paragraph 2).

3. Could the authors comment on the difference in incidence of omicron infection and aHR in the double omicron primed cohorts A vs B (8.9% vs 10.1%)?

Answer: Each cohort in this study was compared to the other two in separate analyses, resulting in slight variations in cohort composition due to the matching process. These differences influenced cohort demographics, follow-up times, and incidence rates across comparisons. For example, as cited by the reviewer, the cumulative incidence in the omicron double-infection cohort differed between the two comparisons (8.9% versus 10.1%) due to variations in cohort composition and follow-up times during the pandemic. This point has now been clarified in the revised manuscript (Discussion, Page 11, Paragraph 4 and Page 12, Paragraph 1).

4. P16, line 338: In the author contributions section, does PVC refer to Dr. Peter Coyle? If so, maybe the author's name misses an initial.

Answer: Yes, indeed. Thank you for spotting this error. It has now been corrected (**Author contributions**).

5. The supplementary text on the matching procedure (page 11, line 206-209) is not clear to me. Is the data from specific individuals from one study arm (single omicron infection) recycled in the second arm (dual omicron infection) for the calculation of the aHR of omicron infection?

Answer: No, the text is intended to explain that for an individual who experienced, for example, two omicron infections, they would be eligible for inclusion in the single omicron infection cohort after their first infection but before their second infection. During this period, they may have been matched and included in the corresponding analysis. However, they would be censored from the single omicron infection cohort upon acquiring the second omicron infection. Following the second infection, the individual would then become eligible for inclusion in the dual omicron infection cohort and could be matched and included in the subsequent analysis. In other words, some individuals could have been matched more than once due to changes in their eligibility for inclusion in different cohorts (a cross-over design). However, no individual contributed to multiple cohorts *simultaneously* under the same inclusion criteria. This point has now been clarified in the revised Supplementary Appendix (**Supplementary Section 5, Page 22, Paragraph 1**).

Referee #3 (Remarks to the Author):

In the present study Abu-Raddad and colleagues conducted an epidemiological study demonstrating the incidence of SARS-CoV-2 omicron (O) infections among individuals with different histories of SARS-CoV-2 immune challenges. In particular, they conducted pairwise comparisons of three groups of individuals: individuals previously infected by O, individuals with two previous O infections, and individuals who got infected by pre-omicron (preO) variant followed by O infection. They also did a subgroup analysis where they further separated and compared individuals that did or did not receive wild-type-(WT)-based vaccine.

Comment: We thank the reviewer for the time and effort put into this review, the assessment of our work, and the constructive feedback on our manuscript that enriched it and improved its readability. Please find below a point-by-point reply addressing each of the reviewer's comments.

The manuscript includes important epidemiological data with a robust grouping approach where individuals are matched 1:1 by various possibly confounding factors. While the observed differences between the groups seem reliable, the interpretation in the context of immunological imprinting is not appropriate for several reasons:

1) Immune imprinting shapes the immune response to infection when the infection is already established and affects the severity of the outcome but is unlikely to impact protection from infection. All these individuals had a previous O infection and have successfully recovered from it. Also, no differences in the severity of previous O infections were reported for different groups. This means that they generated an immune response that is able to

neutralize O (whether this response was imprinted by previous exposures or not) and have a certain degree of protection from reinfection with O (which varies with different immune histories regardless of imprinting).

Answer: We agree with the reviewer that we used a different definition of immune imprinting than the one applied by the reviewer, which contributed to apparently differing interpretations of the findings. Consistent with our previous publications⁹⁻¹², we defined immune imprinting as the broad phenomenon describing how a specific sequence of immunological events—arising from infection and/or vaccination—can either enhance or compromise an individual's future immune protection against both infection and severe disease.

To address this point, we revised the manuscript as follows: First, we explicitly included the definition of immune imprinting at the outset of the paper to clarify the intended meaning of the term, eliminate ambiguity, and provide broader context. Second, we revised the Abstract, Introduction, and Discussion sections to exclude references to immune imprinting in the interpretation of our results. Instead, we focused on the direct observed findings of this study, specifically how different immune histories influence protection against infection. Third, we removed the historical term "antigenic sin" from the manuscript to ensure the focus remains on the observed results of these cohort studies. These revisions have been implemented throughout the Abstract, Introduction, and Discussion sections (**Abstract, Introduction, and Discussion**).

2) Even if imprinting would impact protection from infection its effects should be pronounced in the case of multiple previous exposures to WT and pre-omicron antigens that are structurally more distinct from O antigens than antigens from other O subvariants. Moreover, multiple previous exposures to O antigens should offer better protection against severe O infection, first because the immune response recognizing O is repeatedly boosted and second because there is more chance for overriding the imprinting by WT vaccine and preO variants. The manuscript claims the opposite namely that individuals with WT(vaccine)-O(infection) and those with WT(vaccine)-preO(infection)-O(infection) exposure history are better protected from O infection than individuals with WT(vaccine)-O(infection)-O(infection) exposure history. In the case of the last comparison where unvaccinated individuals with preO(infection)-O(infection) exposure history were found to be better protected than those with only O(infection) this again disproves the role of imprinting since previous exposure to antigenically distinct preO variants should imprint the response to O infection, while O infection cannot imprint response to O infection (if we disregard the genetic differences of omicron subvariants that were also not accounted for in this study). This difference is more likely due to additional immune events and higher antibody titers as many studies demonstrated low adaptive immunity following SARS-CoV-2 infection without previous immune challenges.

Answer: We appreciate the valuable insights provided by the reviewer and would like to clarify that we do not disagree with these points. We believe that differing definitions of immune imprinting have contributed to a misunderstanding of the interpretation of our results. As noted in our response to the first point above, we have now explicitly included the definition of immune imprinting at the outset of the paper to clarify the intended meaning of the term, eliminate ambiguity, and provide broader context. We also revised the Abstract, Introduction,

and Discussion sections to exclude references to immune imprinting in the interpretation of our results. Instead, we focused on the directly observed findings of this study, specifically how different immune histories influence protection against infection. These revisions have been implemented throughout the Abstract, Introduction, and Discussion sections (**Abstract, Introduction, and Discussion**). Additionally, we have included the reviewer's useful insight regarding the interpretation of the results from the third comparison, in terms of additional immune events and higher antibody titers (**Discussion, Page 9, Paragraph 2**).

I have a feeling that the authors do not completely understand the mechanism of immunological imprinting and the difference between immunological imprinting and original antigenic sin. Immunological imprinting occurs when the immune response against a certain pathogen, usually a virus, cannot adapt to the mutations found in a new viral variant but rather remains locked in an initial clonal repertoire imprinted by the previous exposure to a similar yet structurally distinct viral variant. The boosting of adaptive immunity recognizing both variants, however, still occurs and usually offers sufficient protection from new variants. In rare cases, an imprinted immune response can lead to a failure of control over viral replication if a virus mutates to the point where it is still recognized but no longer efficiently neutralized (due to weaker binding of antibodies and TCRs to mutated regions) by the imprinted adaptive immune response. Only then we can talk about the original antigenic sin, and such cases were previously described for influenza and dengue virus infection. Overall, I find the results of this study important but I recommend the authors to interpret their findings in a different way since the proposed immunological imprinting model does not fit.

Answer: We apologize for causing confusion regarding the use of the term and concept of immune imprinting. As noted in our response to the first point above, we have revised the manuscript as follows: First, we explicitly included the definition of immune imprinting at the outset of the paper to clarify the intended meaning of the term, eliminate ambiguity, and provide broader context. Second, we revised the Abstract, Introduction, and Discussion sections to exclude references to immune imprinting in the interpretation of our results. Instead, we focused on the directly observed findings of this study, specifically how different immune histories influence protection against infection. Third, we removed the historical term "antigenic sin" from the manuscript to ensure the focus remains on the observed results of these cohort studies. These revisions have been implemented throughout the Abstract, Introduction, and Discussion sections (**Abstract, Introduction, and Discussion**).

Major comments:

1) How were the SARS-CoV-2 variants determined? By sequencing?

Answer: We have now provided further details on Qatar's variant genomic surveillance program, which includes viral genome sequencing and multiplex real-time reverse-transcription PCR (RT-qPCR) variant screening performed on weekly collected random positive clinical samples^{1,2} (**Methods, Page 16, Paragraph 5 and Page 17, Paragraph 1**). Please also note the description included in Supplementary Section 2 (**Supplementary Section 2**).

2) Wow was the timing between immune challenges for different groups. Particularly

important is to see the average time that passed between the last exposure and the period of observation.

Answer: Thank you for the useful suggestion. This addition has now been included in the Results section (Results, Page 5, Paragraph 3; Page 6, Paragraph 4; and Page 7, Paragraph 4).

3) The differences between groups are observed only in the case of vaccinated individuals for the first two comparisons and only in the case of unvaccinated individuals for the third comparison. How do authors interpret this? Please add to the discussion.

Answer: Since this study provides epidemiological evidence, we are cautious not to overinterpret the findings in terms of basic biological mechanisms. Instead, we focus on the directly observed outcomes of this study, specifically how different immune histories influence protection against infection. As noted in the manuscript (Introduction, Page 4, Paragraph 3), the overarching aim of this specific series of immune history studies⁹⁻¹² is to explore immune history effects at the population level as a basis for hypothesis generation. A detailed explanation of these observed effects is best achieved through controlled basic science experimental studies, both in vitro and in vivo, to elucidate the biological mechanisms underlying the observed population-level effects, as emphasized in this manuscript (Discussion, Page 9, Paragraph 1 and Page 13, Paragraph 2).

Nonetheless, we included the reviewer's useful insight regarding the interpretation of the results from the third comparison, in terms of additional immune events and higher antibody titers (Discussion, Page 9, Paragraph 2). We also believe the findings may be broadly (rather than specifically) interpreted within the framework of the concept that immunity involving both pre-omicron and omicron exposures enhances protection, whereas repeated sequential exposures to the same immunological event reduce protection against new variants, as noted in the manuscript (Abstract, Discussion, Page 9, Paragraphs 2 and 3; and Page 13, Paragraph 2).

Minor comments:

4) Line 53,75: Antigenic sin is a negative term and cannot enhance immune protection. Immune imprinting and antigenic sin are also not synonyms, please look at the definition above.

Answer: As noted in our responses above, we have revised the manuscript as follows: First, we explicitly included the definition of immune imprinting at the outset of the paper to clarify the intended meaning of the term, eliminate ambiguity, and provide broader context. Second, we revised the Abstract, Introduction, and Discussion sections to exclude references to immune imprinting in the interpretation of our results. Instead, we focused on the directly observed findings of this study, specifically how different immune histories influence protection against infection. Third, we removed the historical term "antigenic sin" from the manuscript to ensure the focus remains on the observed results of these cohort studies. These revisions have been implemented throughout the Abstract, Introduction, and Discussion sections (Abstract, Introduction, and Discussion).

5) Line 55: Imprinting effects of what on what?

Answer: As noted in our responses above, we have revised the Abstract, Introduction and Discussion sections to exclude references to immune imprinting in the interpretation of our results. Instead, we focused on the directly observed findings of this study, specifically how different immune histories influence protection against infection. These revisions have been implemented throughout the Abstract, Introduction, and Discussion sections (**Abstract, Introduction, and Discussion**).

6) *Line 67: Hybrid immunity in the context of SARS-CoV-2 infection is usually used to describe immunity by vaccination and infection; this study was not designed to compare that.*

Answer: Thank you for the useful point. We have revised the term 'hybrid immunity' to 'pre-omicron and omicron immunity' throughout the manuscript and Supplementary Appendix to avoid confusion over the use of this term (**Multiple instances throughout the manuscript and Supplementary Appendix**).

7) *The results show that individuals who have fewer past omicron infections are less likely to become infected by omicron again. Might there be a genetic or some other kind of predisposition such as exposure, or a bad immune system, making some individuals more susceptible to omicron infection?*

Answer: This is not likely to be the case. We employed rigorous matching in this study, such as matching by age, coexisting conditions, and nationality, the latter possibly serving as a partial proxy for genetic factors. Moreover, Qatar's population is predominantly young and consists primarily of healthy, working-age migrants, resulting in a very small proportion of elderly individuals who are more likely to have weaker immune systems^{4,13,14}. Despite this, the observed effects were substantial in size by comparison. This clarification has now been included in the revised manuscript (**Discussion, Page 12, Paragraph 3**).

8) *Lines 144-146: This study was designed to only compare differences depending on prior histories of infection and not vaccination (you cannot compare unvaccinated with vaccinated because there is no matching anymore)*

Answer: We apologize for the confusion. We have now clarified that our intention was to convey that the investigated effects differed between the subgroups of vaccinated and unvaccinated individuals (**Discussion, Page 9, Paragraph 1**).

9) *Lines 146-147: Different immune histories impact the protection from O reinfection also without the imprinting.*

Answer: As noted in our responses above, we have revised the Abstract, Introduction and Discussion sections to exclude references to immune imprinting in the interpretation of our results. Instead, we focused on the directly observed findings of this study, specifically how different immune histories influence protection against infection. These revisions have been implemented throughout the Abstract, Introduction, and Discussion sections (**Abstract, Introduction, and Discussion**).

10) Line 146: For clarity, please add that the only omicron-infected group was also unvaccinated.

Answer: We are uncertain about the exact point the reviewer is referring to, as there appears to be a typo in the cited line number (same line number as in the comment right above). However, please note that the Discussion section has been substantially revised to incorporate feedback from all three reviewers, and we hope this revision has addressed this specific point raised by this reviewer (Discussion Section).

11) Were the subvariants or at least the timing of infections comparable between the groups? It would be helpful to add a graph depicting the distribution of infection time points for each group.

Answer: Thank you for the useful suggestion. To address the reviewer's comment, we have added a table showing the distribution of omicron subvariants among the reinfections in each cohort study (Supplementary Table 1) and cited this table in the main text (Results, Page 5, Paragraph 4 and Page 6, Paragraph 5; and Page 7, Paragraph 5). Additionally, we have included a figure illustrating SARS-CoV-2 infection incidence and variants in Qatar from February 5, 2020, the onset of the pandemic, to August 12, 2024, the end of the study (Supplementary Fig. 8).

12) Line 179: Is precluded the right word here?

Answer: This sentence has been revised, and the word in question has been replaced to eliminate confusion (Discussion, Page 10, Paragraph 3).

13) Why the aHR for unratified data is not presented in Figure 3 and why stratified data by vaccination is not presented in Figure 2?

Answer: Thank you for the useful suggestions. In Fig. 3, we aimed to compare the results for the subgroups of vaccinated and unvaccinated individuals; therefore, the unstratified results were not originally included. In response to the reviewer's suggestion, we have now included also the unstratified results as recommended (Fig. 3).

Additionally, we have added a figure showing the cumulative incidence curves of Fig. 2, but now stratified by vaccinated and unvaccinated subgroups, as suggested (Supplementary Fig. 2), and cited this figure in the Results section (Results, Page 6, Paragraph 1; Page 7, Paragraph 1; and Page 8, Paragraph 1).

14) Why are results from Figure 2 not commented on in the discussion?

Answer: The Discussion section focuses on discussing the observed effects, as quantified statistically by the hazard ratios rather than cumulative incidence. However, in response to the reviewer's comment, we have now explicitly cited this figure in the Discussion section when

commenting on the varying incidence based on prior infection histories (Discussion, Page 9, Paragraph 1).

References

- 1 Benslimane, F. M. *et al.* One Year of SARS-CoV-2: Genomic Characterization of COVID-19 Outbreak in Qatar. *Front Cell Infect Microbiol* **11**, 768883, doi:10.3389/fcimb.2021.768883 (2021).
- 2 Hasan, M. R. *et al.* Real-Time SARS-CoV-2 Genotyping by High-Throughput Multiplex PCR Reveals the Epidemiology of the Variants of Concern in Qatar. *Int J Infect Dis* **112**, 52-54, doi:10.1016/j.ijid.2021.09.006 (2021).
- 3 Abu-Raddad, L. J. & Chemaitelly, H. Response to: The imprinting effect of covid-19 vaccines: an expected selection bias in observational studies by S. Monge, R. Pastor-Barriuso, and M.A. Hernan. *BMJ* **381**, e074404, doi:10.1136/bmj-2022-074404 (2023).
- 4 Abu-Raddad, L. J. *et al.* Characterizing the Qatar advanced-phase SARS-CoV-2 epidemic. *Sci Rep* **11**, 6233, doi:10.1038/s41598-021-85428-7 (2021).
- 5 Ayoub, H. H. *et al.* Mathematical modeling of the SARS-CoV-2 epidemic in Qatar and its impact on the national response to COVID-19. *J Glob Health* **11**, 05005, doi:10.7189/jogh.11.05005 (2021).
- 6 Coyle, P. V. *et al.* SARS-CoV-2 seroprevalence in the urban population of Qatar: An analysis of antibody testing on a sample of 112,941 individuals. *iScience* **24**, 102646, doi:10.1016/j.isci.2021.102646 (2021).
- 7 Al-Thani, M. H. *et al.* SARS-CoV-2 Infection Is at Herd Immunity in the Majority Segment of the Population of Qatar. *Open Forum Infect Dis* **8**, ofab221, doi:10.1093/ofid/ofab221 (2021).
- 8 Jeremijenko, A. *et al.* Herd Immunity against Severe Acute Respiratory Syndrome Coronavirus 2 Infection in 10 Communities, Qatar. *Emerg Infect Dis* **27**, 1343-1352, doi:10.3201/eid2705.204365 (2021).
- 9 Chemaitelly, H. *et al.* Immune Imprinting and Protection against Repeat Reinfection with SARS-CoV-2. *N Engl J Med*, doi:10.1056/NEJMc2211055 (2022).
- 10 Chemaitelly, H. *et al.* History of primary-series and booster vaccination and protection against Omicron reinfection. *Sci Adv* **9**, eadh0761, doi:10.1126/sciadv.adh0761 (2023).
- 11 Chemaitelly, H. *et al.* Long-term COVID-19 booster effectiveness by infection history and clinical vulnerability and immune imprinting: a retrospective population-based cohort study. *Lancet Infect Dis* **23**, 816-827, doi:10.1016/S1473-3099(23)00058-0 (2023).
- 12 Chemaitelly, H. *et al.* BNT162b2 Versus mRNA-1273 Vaccines: Comparative Analysis of Long-Term Protection Against SARS-CoV-2 Infection and Severe COVID-19 in Qatar. *Influenza Other Respir Viruses* **18**, e13357, doi:10.1111/irv.13357 (2024).
- 13 Chemaitelly, H. *et al.* Waning of BNT162b2 Vaccine Protection against SARS-CoV-2 Infection in Qatar. *N Engl J Med* **385**, e83, doi:10.1056/NEJMoa2114114 (2021).
- 14 AlNuaimi, A. A. *et al.* All-cause and COVID-19 mortality in Qatar during the COVID-19 pandemic. *BMJ Glob Health* **8**, doi:10.1136/bmjgh-2023-012291 (2023).

Immune Histories and Natural Infection Protection during the Omicron Era: A Population-Based Cohort Analysis

REPLY TO REVIEWERS' COMMENTS

We are grateful to the editor, editorial board, and reviewers for assessing our work and for their insightful and useful feedback and suggestions. Please find below a point-by-point reply addressing each of the comments. We have also incorporated these suggestions in the revised manuscript, as noted below. We would be pleased to address any additional matters, should that be necessary.

Note: All references to the revised manuscript pertain to the marked copies of the manuscript files including changes implemented through "track changes".

Dear Professor Abu-Raddad,

Sincere apologies for the delay in getting back to you with a decision. Your manuscript entitled "Immune Histories and Natural Infection Protection during the Omicron Era: A Population-Based Cohort Analysis" has now been seen by 3 referees. You will see from their comments below that while Reviewers 1 and 3 are happy with the revision and they find your work of considerable interest, some important points are raised still raised by Reviewer 3. We are interested in the possibility of publishing your study in Communications Medicine, but would like to consider your response to these concerns in the form of a revised manuscript before we make a final decision on publication.

We therefore invite you to revise and resubmit your manuscript, taking into account the points raised and addressing them in full. Please highlight all changes in the manuscript text file.

We are committed to providing a fair and constructive peer-review process. Do not hesitate to contact us if you wish to discuss the revision in more detail or if there are specific requests from the reviewers that you believe are technically impossible or unlikely to yield a meaningful outcome.

Comment: We thank the editor for the time and effort put into handling this submission, the assessment of our work, and the constructive feedback on our manuscript that enriched it and improved its readability. Please find below a point-by-point reply addressing each of the editorial and reviewers' comments.

At the same time, we ask that you ensure your manuscript complies with our editorial policies. Please see our revision file checklist for guidance on formatting the manuscript and complying with our policies. You will also find guidelines for replying to the referees' comments.

Communications Medicine seeks to improve the standards and transparency of reporting in our papers, and to ensure that all submissions conform with the editorial policies of Nature

Research. When uploading your revised files please complete and submit Reporting Summary and Editorial Policy checklists as 'checklist' file types. Please note that these forms are a dynamic 'smart pdf' and must therefore be downloaded and completed in Adobe Reader, instead of being opened in a web browser. All points on the checklists must be addressed; if needed, please revise your manuscript in response to these points.

Your revised paper will not be returned to the editors for evaluation until these forms are provided.

Please use the following link to submit your revised manuscript, point-by-point response to the referees' comments (which should be in a separate document to the cover letter), reporting summary, editorial policy checklist and any additional files:

<https://mts-commsmed.nature.com/cgi-bin/main.plex?el=A6DK1BRM6B6fiR3I2A9ftdfUng2yzlg9nLTDerbNTwZ>

*** This url links to your confidential home page and associated information about manuscripts you may have submitted or be reviewing for us. If you wish to forward this email to co-authors, please delete the link to your homepage first ***

We hope to receive your revised manuscript within three months. Please get in touch if you think you might need more time.

Please do not hesitate to contact me if you have any questions or would like to discuss these revisions further. We look forward to seeing the revised manuscript and thank you for the opportunity to review your work.

Answer: We thank the editor for the time and effort put into handling this submission. These instructions have now been addressed.

Referee expertise:

Referee #1: Biostatistics, cohort studies, covid

Referee #2: Antigenic sin, covid, immune responses

Referee #3: Antigenic sin, covid, immune responses

Reviewers' comments:

Referee #1 (Remarks to the Author):

A thorough and effective revision in response to all the reviewer comments.

Comment: We thank the reviewer for the time and effort put into this review, the assessment of our work, and the constructive feedback on our manuscript that enriched it and improved its readability.

Referee #2 (Remarks to the Author):

I thanks the authors for thoroughly addressing my questions, comments and suggestions where possible. Regarding the suggested subgroup analyses on vaccine-primed vs infection-primed individuals, I understand that despite the substantial study cohort sizes, the possibility to perform these analyses were limited and acknowledge the effort. Therefore, I do not have any further questions or comments regarding this manuscript.

Comment: We thank the reviewer for the time and effort put into this review, the assessment of our work, and the constructive feedback on our manuscript that enriched it and improved its readability.

Referee #3 (Remarks to the Author):

The authors have successfully addressed most of my points and the manuscript has improved considerably. However, a few points still need to be addressed:

Comment: We thank the reviewer for the time and effort put into this review, the assessment of our work, and the constructive feedback on our manuscript that enriched it and improved its readability. Please find below a point-by-point reply addressing each of the reviewer's comments.

1) The comment regarding the timing of immune challenges has not been adequately addressed. Please provide the median dates of all infections and vaccinations (or the median duration between immune challenges, but please be consistent). Particularly important is the median time between the last immune challenge and the detected omicron reinfection during the observation period. This may help in the interpretation of your results. It would be helpful if you could graph the timing of the immune challenges (median dates) as a timeline.

Answer: In addition to the revisions made in the previous version of this manuscript, we have now included a detailed figure illustrating the median dates of all immunological events occurring before or after the start of follow-up among individuals who experienced these events in each analysis of this study (**Supplementary Fig. 2**). This figure has also been highlighted in the Results section (**Results Page 5, Paragraph 2; Page 6, Paragraph 3; and Page 7, Paragraph 3**).

2) Line 222-224: Protection against SARS-CoV-2 infection is neither strong nor long-lasting, there is little protection as individuals are re-infected repeatedly, even within a few weeks after vaccination when protection should be at its peak. Even if you wanted to talk about protection against severe infection, I would be cautious about saying that it is strong or durable (if you have found studies that show this, please cite them). Less severe infections are also due to the

virus losing its fitness while adopting mutations to escape immune control.

Answer: Thank you for these insights. We have now revised this statement to incorporate the reviewer's suggestions and have added additional citations (Discussion page 11, Paragraph 1).